# Stochastic Normalization

**Zhi Kou,**[*] **Kaichao You,**[*] **Mingsheng Long** (✉), **Jianmin Wang**
School of Software, BNRist, Research Center for Big Data, Tsinghua University, China
{kz19,ykc20}@mails.tsinghua.edu.cn, {mingsheng,jimwang}@tsinghua.edu.cn

## Abstract

Fine-tuning pre-trained deep networks on a small dataset is an important component in the deep learning pipeline. A critical problem in fine-tuning is how to avoid over-fitting when data are limited. Existing efforts work from two aspects: (1) impose regularization on parameters or features; (2) transfer prior knowledge to fine-tuning by reusing pre-trained parameters. In this paper, we take an alternative approach by refactoring the widely used Batch Normalization (BN) module to mitigate over-fitting. We propose a two-branch design with one branch normalized by mini-batch statistics and the other branch normalized by moving statistics. During training, two branches are stochastically selected to avoid over-depending on some sample statistics, resulting in a strong regularization effect, which we interpret as "architecture regularization." The resulting method is dubbed stochastic normalization (**StochNorm**). With the two-branch architecture, it naturally incorporates pre-trained moving statistics in BN layers during fine-tuning, exploiting more prior knowledge of pre-trained networks. Extensive empirical experiments show that StochNorm is a powerful tool to avoid over-fitting in fine-tuning with small datasets. Besides, StochNorm is readily pluggable in modern CNN backbones. It is complementary to other fine-tuning methods and can work together to achieve stronger regularization effect.

## 1 Introduction

Training deep networks (Szegedy et al., 2015; He et al., 2016b; Huang et al., 2017) from scratch requires large amounts of data. Nevertheless, data collecting is not easy. It took years to build the ImageNet dataset (Deng et al., 2009) with millions of images. For each new task at hand, it is unrealistic to collect a new dataset at the scale of ImageNet. Thanks to the release of pre-trained deep networks in PyTorch (Benoit et al., 2019) and TensorFlow (Abadi et al., 2016), practitioners can benefit from deep learning (LeCun et al., 2015) even with a small amount of data. The practice of transferring pre-trained parameters, a.k.a. *fine-tuning*, is prevalent in both computer vision (Jung et al., 2015) and natural language processing (Devlin et al., 2019).

Because the dataset used in fine-tuning is typically very small, the universal approximation ability (Zhang et al., 2017) of neural networks makes them prone to over-fitting, which is a critical problem in fine-tuning. In general, there are two ways to avoid over-fitting during fine-tuning: impose appropriate regularization to explicitly reduce over-fitting, and transfer prior knowledge by reusing pre-trained networks as an initialization point for implicit regularization.

To avoid over-fitting better, we choose to refactor the widely used Batch Normalization (BN) (Ioffe & Szegedy, 2015) module and come up with a two-branch design: one branch is normalized by mini-batch statistics and the other branch is normalized by moving statistics. During training, the activations of each channel are normalized by either mini-batch statistics or moving statistics, which is determined stochastically in a dropout (Srivastava et al., 2014) style to avoid over-depending on

---

[*]Equal contribution, in alphabetic order

Table 1: Comparing StochNorm and existing methods on regularization type and knowledge transfer.

| method | regularization | layer type | knowledge | transferred |
|---|---|---|---|---|
| L2-SP | parameter | convolutional layer | weight & bias | $\checkmark$ |
| DELTA | feature | batch normalization | weight & bias | $\checkmark$ |
| BSS | feature | batch normalization | moving statistics | $\times$ ($\checkmark$ for StochNorm) |
| *StochNorm* | *architecture* | fully-connected layer | weight & bias | $\times$ |

some sample statistics. The proposed stochastic normalization (**StochNorm**) brings a straightforward byproduct: the moving statistics branch can naturally inherit pre-trained moving statistics which are discarded by existing methods. Therefore, StochNorm transfers more pre-trained knowledge than existing methods to better combat over-fitting.

Table 1(left) lists primary regularization techniques. Li et al. (2018) regularize the parameters near their pre-trained values, Li et al. (2019b) regularize the features near features computed by pre-trained networks, and Chen et al. (2019) penalize small eigenvalues of feature representations. *As an alternative to parameter regularization and feature regularization, the proposed StochNorm regularizes fine-tuning by module design, which we interpret as "architecture regularization."*

Table 1(right) lists whether each type of knowledge is transferred during fine-tuning, with a focus on commonly used ConvNets. Knowledge-free layers like max-pooling and ReLU function are omitted in the table. Usually ConvNets are constructed by stacking Conv-BN-ReLU blocks, followed by a task-specific fully-connected layer. It is a common belief that the knowledge in fully-connected layers is task-specific and cannot be transferred. Transferring learnable parameters (weight and bias) is as easy as just reusing them. Nevertheless, moving statistics in BN layers are simply discarded due to the characteristic behavior of BN (see Section 4.2). *The proposed StochNorm also transfers moving statistics of pre-trained networks to exploit prior knowledge in pre-trained networks better.*

In summary, we study the problem of avoiding over-fitting during fine-tuning, and propose StochNorm, a pluggable module that can be easily in place of BN layers. The novel two-branch architecture design and the stochastic selection mechanism facilitate explicit architecture regularization, while the transfer of pre-trained moving statistics brings implicit initialization regularization, both making StochNorm a powerful tool for fine-tuning with small data. We compare StochNorm with state-of-the-art fine-tuning methods and empirically validate its efficacy with limited data. StochNorm is also complementary to them and they can achieve better performance when combined together.

## 2 Related Work

Our work is related to regularization and normalization techniques used in deep learning, which are reviewed respectively in the following.

### 2.1 Normalization Techniques

Normalizing input features helps optimization because widely used first-order optimization algorithms such as SGD work better on more isotropic landscape (Boyd & Vandenberghe, 2004). Later Ioffe & Szegedy (2015) propose Batch Normalization (BN) to normalize intermediate feature maps by statistics computed with mini-batch samples and find that it greatly helps training of deep networks.

Inspired by Ioffe & Szegedy (2015), many normalization techniques are introduced to deal with different learning scenarios. Layer Normalization (Ba et al., 2016) and Recurrent Batch Normalization (Cooijmans et al., 2017) are effective in recurrent neural networks, Group Normalization (Wu & He, 2018) is designed for object detection, Instance Normalization (Ulyanov et al., 2016) fastens the neural stylization, Weight Normalization (Salimans & Kingma, 2016) speeds up convergence of SGD by a simple re-parameterization, and Spectral Normalization (Miyato et al., 2018) addresses the mode collapse problem in generative adversarial networks. Shekhovtsov & Flach (2018) interprets BN as Bayesian learning, and proposes how to incorporate Bayesian learning into other normalization modules. These normalization modules are tailored to specific optimization problems but are not

related to fine-tuning. Among them, BN is the most widely used normalization module in deep learning. Thus, this paper focuses on ConvNets normalized by BN layers.

BN suffers when the batch-size is small (because of the limited GPU memory) and the estimated statistics are not accurate. Ioffe (2017) propose Batch Renormalization (BRN) for small batch-size learning by combining mini-batch statistics with moving statistics according to a hand-tuned schedule. Guo et al. (2018) normalize features with recent mini-batches and use double-forward to deal with the resulting distribution shift. Peng et al. (2018) utilize distributed training across multiple GPUs to manually increase the batch-size. In fine-tuning, the small batch-size is not a bottleneck problem.

Recent advances in self-supervised learning (He et al., 2020; Chen et al., 2020) show promising results of unsupervised pre-trained representations. One of the techniques they use is to replace BN with ShuffleBN (He et al., 2020), which uses mini-batch statistics of other samples to replace current mini-batch statistics. They confirm that ShuffleBN greatly improves unsupervised contrastive learning. Inspired by this practice, we assume refactoring the BN module can potentially improve the ability of networks to avoid over-fitting during fine-tuning.

Despite its empirical success, BN still lacks a convincing theoretical support on why it works. In the first BN paper, Ioffe & Szegedy (2015) claimed that BN works by reducing the internal covariate shift, which was challenged by several following papers (Bjorck et al., 2018; Santurkar et al., 2018). The proposed StochNorm in this paper randomly mix parameter-sensitive mini-batch statistics with parameter-insensitive moving statistics, effectively improving fine-tuning performance. The success of StochNorm may motivate something new for further research: BN works not by reducing internal covariate-shift, but by allowing for a quick adjustment when a covariate-shift happens.

## 2.2 Regularization Techniques

Regularization is an important topic in machine learning (Bishop, 2006). It is also important for fine-tuning on a small dataset. Existing regularization techniques in fine-tuning can be categorized into parameter regularization and feature regularization, as shown in Table 1(left). Li et al. (2018) regularize the fine-tuned parameters near the pre-trained parameters by L2 constraint, which we call "parameter regularization". Li et al. (2019b) regularize new feature maps near the feature maps computed by pre-trained networks with loss-aware attention and Chen et al. (2019) penalize small eigenvalues of fine-tuned representations, which we call "feature regularization". These methods carefully design additional loss functions to achieve proper regularization.

Besides introducing additional loss to reflect appropriate prior for better regularization, there are other regularization techniques. Dropout (Srivastava et al., 2014) is one of the most famous regularization techniques in deep learning. It randomly removes some neurons during training so that networks do not over-depend on some neurons but pay attention to all of the features. Disout (Tang et al., 2020) is an improved version of Dropout that perturbs neuron outputs rather than drops them. Dropout is simple to implement and can outperform loss-based regularization (such as weight decay). However, Dropout and BN rarely co-occur in deep learning. Li et al. (2019a) studied why Dropout and BN conflict with each other and found that BN and Dropout suffer from variance shift when combined together. An effective way to integrate BN and Dropout is yet unclear.

The proposed StochNorm in this paper improves regularization by new architecture design rather than adding new loss functions. It has two branches normalized by different statistics. Two branches are stochastically selected in the Dropout (Srivastava et al., 2014) style to prevent over-depending on some sample statistics. Hence it endows a *hybrid normalization-regularization effect*. As a byproduct, it seamlessly transfers moving statistics of BN layers in pre-trained networks during fine-tuning.

## 3 Batch Normalization

Here we introduce notations of Batch Normalization (BN) for later sections. BN (Ioffe & Szegedy, 2015) is widely used in common deep models such as Inception Net (Szegedy et al., 2015) and ResNet (He et al., 2016b). These models usually serve as the *backbones* for fine-tuning tasks.

As a normalization layer, BN takes the input feature map and outputs the normalized feature map. Since BN is a channel-wise normalization technique, we omit the indexing for channel dimension and only discuss the batch dimension in the following sections.

Given a feature map distribution $x \sim P$, BN aims to normalize the feature map by population-level mean and variance: $\widehat{x} = \frac{x - \mathrm{E}_{x \sim P}[x]}{\sqrt{\mathrm{Var}_{x \sim P}[x] + \epsilon}}$, where $\epsilon$ is a small constant to avoid zero division.

Population-level statistics cannot be explicitly computed, but can be estimated by samples $\{x_i\}_{i=1}^{n}$, with $n$ as the size of the dataset. In mini-batch $\{x_i\}_{i=1}^{m}$ with batch size of $m$ ($m \ll n$), BN estimates the mini-batch statistics (mean and variance) by $\mu = \frac{1}{m} \sum_{i=1}^{m} x_i, \sigma^2 = \frac{1}{m} \sum_{i=1}^{m} (x_i - \mu)^2$.

These estimated mini-batch statistics are used to normalize the layer output to have zero mean and unit variance at training stage with $\widehat{x}_i = \frac{x_i - \mu}{\sqrt{\sigma^2 + \epsilon}}$. At the inference stage, mini-batch estimations $\mu$ and $\sigma^2$ are not available, so BN tracks moving average of the statistics during training: $\tilde{\mu} \triangleq \alpha \mu + (1 - \alpha) \tilde{\mu}$, $\tilde{\sigma}^2 \triangleq \alpha \sigma^2 + (1 - \alpha) \tilde{\sigma}^2$, where $\alpha$ is the coefficient of moving average, $\tilde{\mu}$ and $\tilde{\sigma}^2$ are moving average versions of $\mu$ and $\sigma^2$. These moving statistics $\tilde{\mu}$ and $\tilde{\sigma}^2$ are used to normalize the feature map as $\widehat{x}_i = \frac{x_i - \tilde{\mu}}{\sqrt{\tilde{\sigma}^2 + \epsilon}}$ for inference. Note that moving statistics $\tilde{\mu}$ and $\tilde{\sigma}^2$ have no influence on training but only affects inference, which we will revisit in Section 4.2.

To recover the representation power of deep features, BN introduces additional trainable parameters $\beta$ and $\gamma$ to scale and shift the normalized values by $y_i = \gamma \widehat{x}_i + \beta$.

# 4 Stochastic Normalization

## 4.1 Two-Branch Design

During training, the output of BN depends on the statistics of mini-batch data, while the moving statistics $\tilde{\mu}$ and $\tilde{\sigma}^2$ are only used at inference stage. It means networks trained by BN may over-depend on some mini-batch statistics during training, but generalize worse during inference, especially when the number of samples is small and the diversity of mini-batches is low. Hence, we split the normalization process into two branches, one uses mean and variance of current mini-batch data as usual, while the other uses current moving statistics $\tilde{\mu}$ and $\tilde{\sigma}$ of the training data:

$$\widehat{x}_{i,0} = \frac{x_i - \tilde{\mu}}{\sqrt{\tilde{\sigma}^2 + \epsilon}}, \quad \widehat{x}_{i,1} = \frac{x_i - \mu}{\sqrt{\sigma^2 + \epsilon}}. \quad (1)$$

Now that we have two normalization branches, we have to decide which one to use. We propose to stochastically select one branch with probability $p$ for forward propagation in each channel of the normalization layers and each step of training. Let $s$ be the branch-selection variable, then $s$ is generated from the *Bernoulli distribution* $P(s) = \begin{cases} p, & s = 1 \\ 1 - p, & s = 0 \end{cases}$. The discrete selection operator can be written as a summation $\widehat{x}_i = (1 - s)\widehat{x}_{i,0} + s\widehat{x}_{i,1}$. Then the scale and shift parameters $\beta$ and $\gamma$ can be applied after the stochastic forward as usual. By adjusting the selection parameter $p$ of Bernoulli distribution, we can balance the selection of two branches.

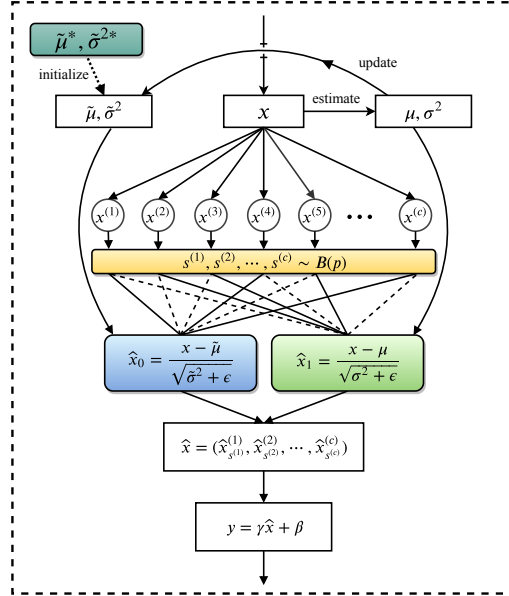

Figure 1: The computation graph of StochNorm at training. $c$ is channel size of input feature map $x$ and $j$ is the channel index; $(\tilde{\mu}^*, \tilde{\sigma}^{2^*})$ are the pretrained moving mean and variance. At inference, the yellow and green parts will be removed.

Note that when $s = 1$, StochNorm boils down to BN. As the dataset in fine-tuning is pretty small, regular training is prone to over-fitting. When $s = 0$, StochNorm uses moving statistics to normalize the activations, which forces the network to learn well even when some statistics deviate from mini-batch statistics. Note that many works (Ioffe & Szegedy, 2015; Ioffe, 2017) have found normalizing the whole feature map using moving statistics can lead to collapse during training. StochNorm

does not suffer from that because only some randomly selected channels are normalized by moving statistics, successfully stabilizing fine-tuning without relying on techniques like BRN. By randomly selecting the value of $s$, we also require the network not to over-depend on some sample statistics. This regularizes the network in a dropout style to combat over-fitting.

StochNorm is intuitively described in Figure 1 and summarized in detail by Algorithm 1.

---

**Algorithm 1** Stochastic Normalization (StochNorm)

---

**Input**: mini-batch feature maps of each channel $x = \{x_i\}_{i=1}^m$;
   moving statistics update rate $\alpha \in (0, 1)$ and learnable parameters $\beta, \gamma$;
   moving statistics $\tilde{\mu}, \tilde{\sigma}^2$ (initialized by $\tilde{\mu}^*, \tilde{\sigma}^{2^*}$) and branch-selection probability $p \in (0, 1)$.
**Output:** $y = \text{StochNorm}(x)$.
**Training:**

$$\mu \leftarrow \frac{1}{m} \sum_{i=1}^m x_i, \sigma^2 \leftarrow \frac{1}{m} \sum_{i=1}^m (x_i - \mu)^2 \qquad \text{// mini-batch mean and variance}$$

$$\widehat{x}_{i,0} \leftarrow \frac{x_i - \tilde{\mu}}{\sqrt{\tilde{\sigma}^2 + \epsilon}}, \quad \widehat{x}_{i,1} \leftarrow \frac{x_i - \mu}{\sqrt{\sigma^2 + \epsilon}} \qquad \text{// normalize with moving/mini-batch statistics}$$
$$\widehat{x}_i = (1-s)\widehat{x}_{i,0} + s\widehat{x}_{i,1}, \ s \sim B(p) \qquad \text{// stochastic forward-prop for each channel}$$

$$y_i \leftarrow \gamma \widehat{x}_i + \beta \qquad \text{// scale and shift}$$

$$\tilde{\mu} \leftarrow \tilde{\mu} + \alpha(\mu - \tilde{\mu}), \tilde{\sigma}^2 \leftarrow \tilde{\sigma}^2 + \alpha(\sigma^2 - \tilde{\sigma}^2) \qquad \text{// update estimations of moving statistics}$$

**Inference:**

$$y_i \leftarrow \gamma \frac{x_i - \tilde{\mu}}{\sqrt{\tilde{\sigma}^2 + \epsilon}} + \beta \qquad \text{// normalize with moving statistics}$$

---

## 4.2 Transferring Pre-trained Moving Statistics

From the formal notation of BN described in Section 3, it is clear that moving statistics do not affect training. Because the moving statistics take the form of exponential moving average, the influence of initialization of moving statistics decays exponentially. Put it together, pre-trained moving statistics in BN have no influence on fine-tuning.

Nevertheless, moving statistics are computed by data and parameters which are learned from data, so moving statistics in pre-trained BN layers also have valuable knowledge about pre-trained data. Because StochNorm has a branch which normalizes activations by moving statistics, it is natural to initialize the moving statistics as the pre-trained moving statistics $\tilde{\mu}^*$ and $\tilde{\sigma}^{2^*}$, as indicated by the top-left part of Figure 1. This way, pre-trained moving statistics join the training through StochNorm and contribute to fine-tuning. This initialization strategy takes effect as an implicit regularization.

In a summary, StochNorm makes fine-tuning robust to over-fitting from two aspects: (i) the two-branch design and the stochastic selection mechanism facilitate a hybrid normalization-regularization effect, which penalizes over-fitting, (ii) more knowledge from pre-trained models is transferred.

## 4.3 Fine-Tuning with StochNorm

**Integration with fine-tuning methods.** Many methods for fine-tuning can be described by a uniform framework with supervised cross-entropy loss function combined with regularization terms. They vary on how to design regularization terms, i.e., how to reuse the knowledge in the pre-trained network while avoiding over-fitting on the small target dataset. They are complementary to StochNorm. By replacing BN with StochNorm in their network backbones, these methods enjoy additional performance gains, as shown in our experiments.

**Integration with network backbones.** As a general and lightweight normalization layer, StochNorm can be easily plugged into common network backbones in place of the standard BN. As shown in Figure 2, we can apply StochNorm to several kinds of deep neural networks, from VGG (Simonyan & Zisserman, 2015), ResNet (He et al., 2016a) to Inception Net (Szegedy et al., 2016), with no other modifications to the network architectures.

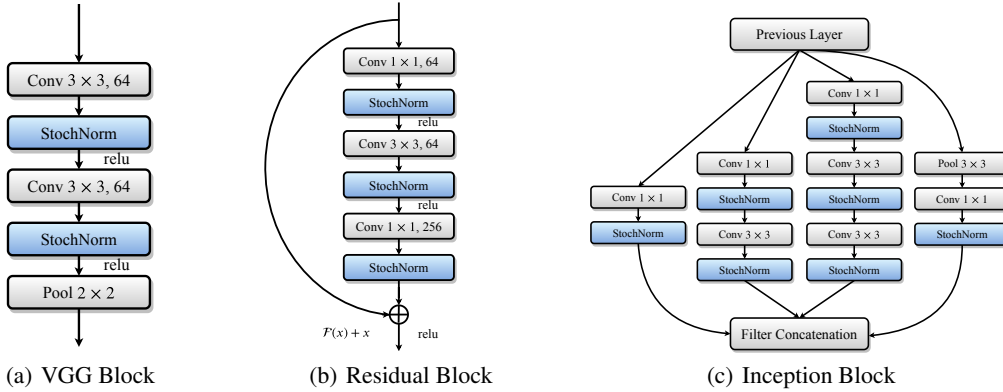

|   (a) VGG Block   |   (b) Residual Block   |   (c) Inception Block   |

Figure 2: Integration of StochNorm in mainstream deep backbones, by replacing BN with StochNorm.

# 5 Experiments

To evaluate StochNorm, we apply it to four visual recognition tasks. Experimental results indicate that our method can outperform state-of-the-art fine-tuning methods over four datasets when there are limited data. We also conduct insight analysis and ablation study to better understand StochNorm. The code is available at `https://github.com/thuml/StochNorm`.

## 5.1 Setup

**Datasets.** The evaluation is conducted on four standard datasets. **CUB-200-2011** (Welinder et al., 2010) is a dataset for fine-grained bird recognition with 200 bird species and $11,788$ images. It is an extended version of the CUB-200 dataset. **Stanford Cars** (Krause et al., 2013) contain $16,185$ images for 196 classes of cars. **FGVC Aircraft** (Maji et al., 2013) is a benchmark for the fine-grained aircraft categorization. The dataset contains $10,000$ aircraft images, with 100 images for each of the 100 categories. **NIH Chest X-ray** (Wang et al., 2017) consists of $112,120$ frontal-view X-ray images of $30,805$ patients with fourteen disease labels (each image can have multiple labels).

**Compared methods.** StochNorm is compared with several fine-tuning methods: vanilla fine-tuning; $\mathbf{L^2}$-**SP** (Li et al., 2018) which regularizes the weight parameters around pre-trained parameters to alleviate catastrophic forgetting; **DELTA** (Li et al., 2019b) which selects features with a supervised attention mechanism; and **BSS** (Chen et al., 2019) which penalize small eigenvalues of feature representations to protect training from negative transfer. Despite of the vanilla fine-tuning, $\mathbf{L^2}$-**SP**, **DELTA**, and **BSS** are state-of-the-art methods to avoid over-fitting in fine-tuning.

**Implementation details.** Experiments are implemented based on PyTorch (Benoit et al., 2019). We follow previous protocols (Li et al., 2018, 2019b; Chen et al., 2019) to train the last fully connected layer from scratch, and to fine-tune the backbone network. The learning rate for the last layer is set to be 10 times of those for the fine-tuned layers because parameters in the last layer are randomly initialized. We adopt SGD with momentum of 0.9 together with the progressive training strategies in Li et al. (2018). Experiments are repeated five times to get the mean and deviation. Hyper-parameters for each method are selected on validation data. We follow the train/validation/test partition of each dataset. For datasets without validation data, we use $20\%$ training data for validation and use the same validation data for all methods. The selection probability $p = 0.5$ works well for most experiments.

## 5.2 Results

**Medical image analysis.** With the rapid development of transfer learning, medical image analysis can benefit from deep learning even with a small amount of data. The **NIH Chest X-ray** dataset consists of X-ray images collected across a whole country and thus is a large-scale dataset. Typically we cannot afford the cost to build such a large dataset. Therefore, we design experiments with $5\%$ samples, $10\%$ samples, and $15\%$ samples. The task in this dataset is multi-label binary classification and the evaluation metric is the average AUC for the fourteen diseases. ResNet-50 (He et al., 2016a) pre-trained on ImageNet serves as the network backbone. Results are summarized in Table 2.

Table 2: Average performances (AUC) of diagnosing different pathologies on NIH Chest X-ray.

| Method | 5% samples | 10% samples | 15% samples |
|---|---|---|---|
| vanilla (baseline) | 70.37±0.31 | 75.85±0.22 | 76.64±0.11 |
| $L^2$-SP | 70.02±0.27 | 72.73±0.41 | 75.71±0.22 |
| DELTA | 70.99±0.19 | 74.35±0.20 | 75.97±0.17 |
| BSS | 69.86±0.12 | 73.27±0.19 | 76.10±0.23 |
| **StochNorm** | **72.50±0.26** | **76.48±0.15** | **77.01±0.21** |

Due to the large discrepancy between medical images and common visual recognition images, compared methods are inferior to vanilla fine-tuning, a phenomenon named negative transfer (Chen et al., 2019). In contrast, the proposed **StochNorm** can achieve an average improvement of **1.1%** across three sampling rates in terms of AUC, by simply replacing BN layers with StochNorm. Although Raghu et al. (2019) have found that vanilla fine-tuning with ImageNet pre-trained models fails to achieve significant improvement on medical imaging dataset, we find that when target dataset is insufficient, there is still room for improvement.

**Fine-grained tasks.** We present the classification results for three fine-grained tasks in Table 3. Note that the bold results are the best results.To explore the effect of StochNorm when the target dataset is small, we randomly sample a proportion of the data to reduce the dataset size. The sampling rates are 15%, 30%, 50% and 100%. Again the network backbone is ResNet-50 pre-trained on ImageNet.

Table 3: Top-1 Accuracy (%) of StochNorm and different methods (Backbone: ResNet-50).

| Dataset | Method | Sampling Rates | | | |
|---|---|---|---|---|---|
| | | 15% | 30% | 50% | 100% |
| CUB-200-2011 | vanilla (baseline) | 45.25±0.12 | 59.68±0.21 | 70.12±0.29 | 78.01±0.16 |
| | $L^2$-SP (Li et al., 2018) | 45.08±0.19 | 57.78±0.24 | 69.47±0.29 | 78.44±0.17 |
| | DELTA (Li et al., 2019b) | 46.83±0.21 | 60.37±0.25 | 71.38±0.20 | 78.63±0.18 |
| | BSS (Chen et al., 2019) | 47.74±0.23 | 62.03±0.29 | **72.56±0.17** | 78.85±0.31 |
| | **StochNorm** | **50.14±0.19** | **62.34±0.26** | 72.01±0.15 | **79.58±0.13** |
| Stanford Cars | vanilla (baseline) | 36.77±0.12 | 60.63±0.18 | 75.10±0.21 | 87.20±0.19 |
| | $L^2$-SP (Li et al., 2018) | 36.10±0.30 | 60.30±0.28 | 75.48±0.22 | 86.58±0.26 |
| | DELTA (Li et al., 2019b) | 39.37±0.34 | 63.28±0.27 | 76.53±0.24 | 86.32±0.20 |
| | BSS (Chen et al., 2019) | 40.57±0.12 | 64.13±0.18 | 76.78±0.21 | **87.63±0.27** |
| | **StochNorm** | **41.08±0.17** | **65.02±0.21** | **77.39±0.26** | 87.35±0.22 |
| FGVC Aircraft | vanilla (baseline) | 39.57±0.20 | 57.46±0.12 | 67.93±0.28 | 81.13±0.21 |
| | $L^2$-SP (Li et al., 2018) | 39.27±0.24 | 57.12±0.27 | 67.46±0.26 | 80.98±0.29 |
| | DELTA (Li et al., 2019b) | 42.16±0.21 | 58.60±0.29 | 68.51±0.25 | 80.44±0.20 |
| | BSS (Chen et al., 2019) | 40.41±0.12 | 59.23±0.31 | **69.19±0.13** | 81.48±0.18 |
| | **StochNorm** | **42.63±0.18** | **60.09±0.25** | 69.00±0.16 | **81.65±0.14** |

As shown in Table 3, StochNorm significantly outperforms vanilla fine-tuning when target data is small (with sampling rate of 15% and 30%). In the challenging setting where there are only 15% data for training, fine-tuning is easily prone to over-fitting. In this case, StochNorm gets an average of **4.1%** increase compared with vanilla fine-tuning, demonstrating the regularization effect of StochNorm. Compared with state-of-the-art fine-tuning methods, StochNorm is superior across a wide spectrum of sampling rates for these three datasets.

It is worth to note that we work on avoiding over-fitting in fine-tuning. It is expected that regularization helps less when more data are available. If there are abundant data (with sampling rate of 50% and 100%), StochNorm sometimes can improve over vanilla fine-tuning but sometimes not. State-of-the-art fine-tuning methods do not show significant improvement with 100% data, either.

## 5.3 Integration with Network Backbones and Other Methods

**Network backbones.** In the above experiments, we adopt the widely used ResNet-50 network backbone. But our method can be easily integrated with other backbones, as described in Section 4.3. Figure 3 shows the absolute improvements in the CUB-200-2011 dataset over vanilla fine-tuning by integrating StochNorm with another two backbones: VGG-16 (Simonyan & Zisserman, 2015) architecture which gradually increases the channel size but decreases the feature map size of convolutional layers; Inception-v3 (Szegedy et al., 2016) architecture which uses parallel convolutional layers with different kernel sizes to achieve multi-scale convolution. StochNorm works well with different backbones and significantly outperforms the vanilla fine-tuning when data are limited, demonstrating that the proposed StochNorm can be easily plugged into popular network backbones for better regularization during fine-tuning.

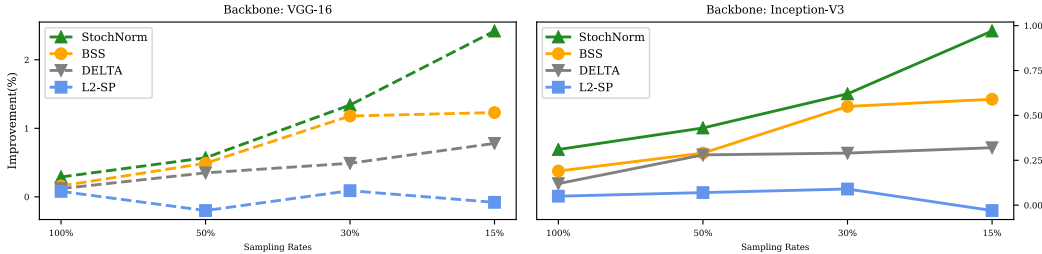

Figure 3: StochNorm with different backbones.

**Integration with other methods**. Here we integrate StochNorm with other regularization methods ($\mathbf{L^2}$-**SP**, **DELTA** and **BSS**) for fine-tuning. Table 4 summarizes the results. It is clear that integrating StochNorm with other methods can improve their performance. This is because StochNorm is complementary to them. While $\mathbf{L^2}$-**SP** imposes parameter regularization and **DELTA** and **BSS** impose feature regularization, the proposed StochNorm introduces architecture regularization by designing a better module for fine-tuning.

Table 4: Accuracy of StochNorm integrated with different methods (Backbone: ResNet-50).

| Method | CUB-200-2011 | | | | FGVC Aircraft | | | | Stanford Cars | | | |
|---|---|---|---|---|---|---|---|---|---|---|---|---|
| | 15% | 30% | 50% | 100% | 15% | 30% | 50% | 100% | 15% | 30% | 50% | 100% |
| $L^2$-SP | 45.08 | 57.78 | 69.47 | 78.44 | 39.27 | 57.12 | 67.46 | 80.98 | 36.10 | 60.30 | 75.48 | 86.58 |
| **+StochNorm** | 49.92 | 60.48 | 70.47 | 79.24 | 42.57 | 60.16 | 69.16 | 81.12 | 40.50 | 64.86 | 77.34 | 86.81 |
| DELTA | 46.83 | 60.37 | 71.38 | 78.63 | 42.16 | 58.60 | 68.51 | 80.44 | 39.37 | 63.28 | 76.53 | 86.32 |
| **+StochNorm** | 49.27 | 62.86 | 72.78 | 79.72 | 44.10 | 60.13 | 70.12 | 81.03 | 40.77 | 65.67 | 77.23 | 86.51 |
| BSS | 47.74 | 62.03 | 72.56 | 78.85 | 40.41 | 59.23 | 69.19 | 81.48 | 40.57 | 64.13 | 76.78 | 87.63 |
| **+StochNorm** | 50.67 | 64.10 | 73.01 | 79.91 | 43.89 | 60.25 | 69.41 | 81.50 | 44.04 | 66.28 | 78.03 | 87.85 |

## 5.4 Ablation study

We conduct ablation study in three fine-grained classification tasks with $15\%$ data to explore each component in StochNorm. Results are presented in Figure 4.

**Initialization of $\tilde{\mu}$ and $\tilde{\sigma}^2$.** Figure 4(a) shows variants of StochNorm with different initialization of $\tilde{\mu}$ and $\tilde{\sigma}^2$. (i) We design an experiment to compare BN with StochNorm whose moving statistics are initialized by $0$ and $1$. Note that StochNorm with (0,1) significantly surpasses BN, confirming the regularization effect of our two-branch design. (ii) We design experiments to explore the benefit of transferring pre-trained moving statistics. Apart from $(0, 1)$ initialization for the moving statistics, we also compare $(\tilde{\mu}_t, \tilde{\sigma}_t^2)$, which are computed by target training data before fine-tuning starts. The proposed StochNorm is denoted by $(\tilde{\mu}^*, \tilde{\sigma}^{2^*})$, meaning that pre-trained moving statistics are used for initialization of the moving statistics. Among these variants, $(\tilde{\mu}^*, \tilde{\sigma}^{2^*})$ is the best, while $(0, 1)$ is the worst. Results indicate that reusing pre-trained moving statistics is beneficial to fine-tuning, although the improvement is not as large as the two-branch design.

**Impact of** $\alpha$**.** Although the hyper-parameter $\alpha$ in Algorithm 1 is actually a hyper-parameter of BN rather than of StochNorm, it now affects the learning (back-propagation). Therefore we compare $\alpha = 0.001$ with PyTorch's default $\alpha = 0.1$ in Figure 4(b), where the y-axis means the absolute improvement of StochNorm over vanilla fine-tuning. Empirically we find that smaller $\alpha$ leads to slower convergence and slightly worse results. The default value $\alpha = 0.1$ works well.

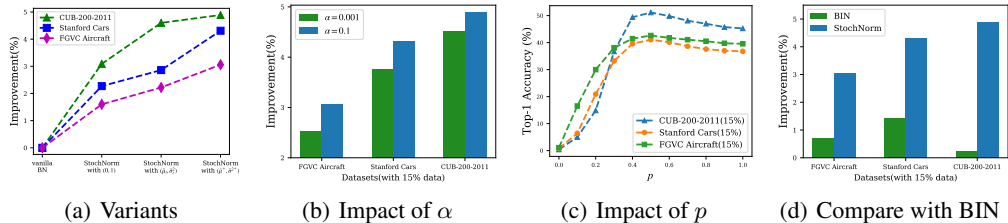

| (a) Variants | (b) Impact of $\alpha$ | (c) Impact of $p$ | (d) Compare with BIN |

Figure 4: Ablation Study.

**Impact of** $p$**.** Figure 4(c) presents an ablation study of $p$. The ablation study confirms our empirical finding that the best $p$ sits around $0.5$ and $p = 0.5$ works well for most experiments. Note that $p = 1$ is the same as BN, while $p = 0$ is to normalize all features with moving statistics, which can lead to collapse during training, as confirmed by many researchers like Ioffe & Szegedy (2015). In StochNorm, only some randomly selected channels are normalized by moving statistics while others by mini-batch statistics, successfully stabilizing fine-tuning without relying on techniques like BRN.

**Comparison with other normalization methods.** Since StochNorm is a normalization module, it is necessary to compare StochNorm with other normalization methods. However, we focus on fine-tuning in this paper and public available pre-trained models are all trained with BN. Therefore, we cannot directly compare StochNorm with normalization techniques like instance normalization (IN) or weight normalization (WN). Batch-Instance Normalization (BIN) can be compared because it is based on BatchNorm. Figure 4(d) presents the comparison with respect to the absolute improvements over BN, which shows that BIN is slightly better than BN but substantially inferior to StochNorm.

## 6 Conclusion

How to alleviate over-fitting in fine-tuning with small datasets is an important problem. This paper proposes Stochastic Normalization (StochNorm) to improve the widely used batch normalization module in a dropout-like way to battle against over-fitting. The two-branch design and transfer of pre-trained moving statistics are empirically confirmed to be helpful for fine-tuning. StochNorm can be easily integrated into popular ConvNets, has top fine-tuning results, and can work with other fine-tuning methods.

## Broader Impact

This paper proposes a new network module called StochNorm as the basic building block of deep neural networks. It can greatly improve fine-tuning of pretrained models in the small data regime. Its broader impact depends on the usage scenario of fine-tuning in deep learning applications. In addition, it may inspire some researchers for further investigation of regularization techniques.

## Acknowledgments and Disclosure of Funding

This work was supported by the National Natural Science Foundation of China (61772299, 71690231), Beijing Nova Program (Z201100006820041), University S&T Innovation Plan by the Ministry of Education of China. We thank Yuchen Zhang for helpful discussions. We also thank reviewers and meta-reviewers for their constructive reviews.

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
