[Reviews · NeurIPS 2020]

Review 1

Summary and Contributions: This paper introduces a novel method to prevent overfitting when fine-tuning a pre-trained network for a new task using a small training set. The paper proposes a hybrid batch normalization layer, called stochastic normalization that, randomly switches the normalization statistics between: those calculated from the current min-batch and the moving average statistics. The authors replace the standard batch normalization layer of different network architectures such as VGG-16, Inception-V3, and Resnet-50 with their proposed stochastic normalization and show empirically that the fine-tuning using the adopted architecture outperforms multiple existing methods for over-fitting problem in fine-tuning. Overall, the paper is studying a very important problem and the proposed method seems to be working in practice. The major problem I have with this paper is the lack of consistency in the experimental set up. The authors cherry-pick baselines or base network backbones for different parts of the paper without any good justification. For example: 1- why other baselines are not reported in Table 2. 2- Why not reporting the results of other baselines for the other network backbones in Figure 3.a? It is not clear whether the method outperforms the state of the art methods on these network backbones. What about the results on the other data sets? 3- Why using different data sets for different experiments? For example, the authors used CUB-200-2011 for network backbones (Figure 3.a) and Stanford Cars for integration study (Table 4). This raises the question whether the authors picked specific data sets for which their proposed method worked better. I am not convinced of authors response to my questions. Note that they are reporting the average across three data sets for Figure (3).a which is a weird setup. Also the values reported for this figure in the original submission do not match the one reported in the rebuttal. Also, the authors failed to answer my other questions. And here are some minor problems: There is a space in numbers in Section 5. Remove them for better readability: 11, 788 --> 11,788 Only for the Stanford Cars, the information for training/test split is provided. Why? After rebuttal: I am not convinced of authors response to my questions. Note that they are reporting the average across three data sets for Figure (3).a now which is a weird setup. Also the values reported for this figure in the original submission do not match the one reported in the rebuttal. Moreover, they did not answer all my questions. So, the rebuttal does not change my evaluation of the paper.

Strengths: Addresses a very important and open problem Extensive Empirical evaluation Is novel and relevant

Weaknesses: Inconsistency in empirical evaluation

Correctness: The claims cannot be verified based on the current empirical methodology

Clarity: Yes, the paper is very clear and easy to follow.

Relation to Prior Work: Yes, it does.

Reproducibility: Yes

Additional Feedback:


Review 2

Summary and Contributions: The paper introduces stochastic normalization (SN), an alternative to improve the generalization of batch normalization (BN) by randomly switching during training between batch statistics and the accumulated statistics. The hypothesis is that training may over-depend on some mini-batch statistics, which are not directly used during inference. By using the moving statistics randomly, the authors expect to improve overfitting. In addition, during fine-tuning on a new dataset after pre-training, the moving statistics computed on the pre-training data can also be transferred to the new dataset.

Strengths: The paper shows that SN improves the results of vanilla BN and other regularization techniques (L2-SP, DELTA, BSS) when transferring from ImageNet to different datasets (CUB, Stanford Cars, FGVC Aircraft, NIH Chest X-ray) and across different backbone architectures (VGG, Inception, ResNet), especially on few data regimes. In addition, StochNorm is orthogonal to the aforementioned regularization techniques and can be employed together with them, improving their results, as shown in Table 4.

Weaknesses: The main criticism is that the method only really shines on low data regimes. For instance, Table 3 shows that BSS performs very closely than the proposed method when using 100% of the available training data for fine-tuning. The proposed method is complementary to BSS and when combined with it, improves the results as shown in Table 4 (only slightly on full data regimes). However, I do not consider this a major weakness. In addition, some results and details could be better presented (see following sections in the review).

Correctness: The method has been properly evaluated through multiple experiments on different datasets, and using different backbone architectures, as well as running the experiments multiple times (5 runs), reporting the standard deviation on accuracy. An ablation study is also conducted to understand how different pieces of SN contribute to the final performance during fine-tuning (i.e. how the mu and sigma parameters are initialized). On line 105-106, the statement "Dropout and BN are mutually exclusive in deep learning, because Dropout affects how BN computes the statistics for normalizing activations." seems a little bit too strong, since this depends on the order in which BN and Dropout are applied. That statement could be true for something like Conv+Dropout+BN, but it's not necessarily true for Dropout+Conv+BN, especially when considering that modern neural networks also include residual connections in many cases. I would suggest to tone down that statement.

Clarity: The paper is clearly written and easy to read. There are only a few minor typos (Line 20, "takes" -> "took"). The most important detail which is missing is the value of the hyperparameter used during the moving statistics computation (usually called beta). This hyperparameter has an enormous effect when fine-tuning to a new dataset, specially if very few steps are performed. If beta is very large (e.g. 0.999) the moving statistics may not have time to be updated during fine-tuning. In general, a section studying how the results change according to this hyperparameter would be appreciated, since it also affects now the optimization of the parameters. In Table 4, it would be better to also add the results of StochNorm alone, so that the reader doesn't need to go back to Table 3 to check the results of StochNorm on Stanford Cars. Regarding the same table, is not clear why the authors decided to perform that experiment on Stanford Cars rather than the other datasets. It would be appreciated if this was clarified. Some minor things that should be clarified: - In the tables and figures captions, describe that the results are the average across 5 runs and what the +/- numbers are (standard deviation). - Describe how sigma and mu in SN are initialized during backbone training (I assume that sigma=1 and mu=0).

Relation to Prior Work: The authors discuss relevant prior work in Section 2, discussing several normalization techniques alternative to BN (such as GroupNorm, LayerNorm, etc), as well as other regularization techniques (such as Dropupt, L2-SP, and others compared in the paper).

Reproducibility: Yes

Additional Feedback: ------ Comments after rebuttal ------ I appreciate the work made by the authors to address many of my comments and those of other reviewers. In particular, I'm glad to see that they experimented with different values of the BN momentum hyperparameter, as well as agreed to tone down some of their claims. I also appreciated the plot exploring different values of $p$ and the additional tables with results on all the datasets. However, I will not increase my score further, since some of my concerns (and some from other reviewers were not addressed in the rebuttal (e.g. small returns with many supervised data).


Review 3

Summary and Contributions: This work presents a two-branch batch normalization. One branch is normalized by the current mini-batch statistics and the other branch is normalized by the historical moving statistics. And then two branches are stochastically selected to avoid overfitting when fine-tuning pre-trained networks on a new small dataset.

Strengths: With the two-branch architecture, it can naturally utilize prior knowledge of pre-trained networks during fine-tuning by stochastically selecting pre-trained moving statistics in batch normalization layers.

Weaknesses: 1) In my experience, it is commonly collapsed when training the network with BN directly normalized by moving statistics. Batch renormalization technique [1] can address this problem. If you have used batch renormalization in the experiments, it is better to cite it in the experimental part. If you used a new technique to avoid training from collapse, please give the details. [1] Ioffe,S. Batch renormalization: Towards reducing minibatch dependence in batch-normalized models. In Advances in neural information processing systems, pp. 1945–1953, 2017. 2) I think it might need more comparison/ablation experiments to show the characteristics of StochNorm. There are no experiments of comparing stochastic normalization to each branch batch normalization with the current mini-batch statistics or the moving statistics. The parameter selection probability $p$ is significant, and it is better to offer the parameter sensitivity analysis in the experiments. 3) StochNorm is orthogonal to other fine-tuning methods, which is claimed to be an advantage. However, when observing Table 3 and Table 4, I find the performances of L^2_SP + StochNorm and DETAL + StochNorm on dataset Stanford Cars is even inferior to that of single StochNorm, which might be inconsistent with the claim. 4) A typical NeurIPS paper commonly offers us insights with theoretically grounded analysis. This work is relatively weak in this aspect.

Correctness: Yes

Clarity: Yes

Relation to Prior Work: Yes

Reproducibility: Yes

Additional Feedback:


Review 4

Summary and Contributions: The authors propose a novel regularization strategy by refactoring Batch Normalization (BN). Their proposal (StochNorm) is focused on alleviating over-fitting in fine-tuning with small datasets. As clearly stated by the authors, "The major contribution of StochNorm is the two-branch design with stochastic selection."

Strengths: The paper is clearly written, well structured, and the topic tackled is very relevant for the community.

Weaknesses: Even if the experimental comparison seems reasonable, and the results are very promising and superior to competitors, I'd like to see more regularizers/normalizers in the comparison. I mean: when I was reading the paper, I was expecting to see the comparison with conventional BN, instance normalization, weight normalization, batch-instance normalization, dropout, and/or techniques of this kind. Also, a comparison between this approach and performing data augmentation would be of great help to see which strategy is more convenient to improve generalization. All with the objective to elucidate which regularization/normalization approach is better. However, the experimental section did not include this type of experiment.

Correctness: Yes, I think so.

Clarity: Yes, it is.

Relation to Prior Work: Yes, I think the paper is sufficiently clear in this regard.

Reproducibility: Yes

Additional Feedback: As already indicated before, when I was reading the paper, I was expecting to see the comparison, using the exact same backbone ConvNet, with conventional BN, instance normalization, weight normalization, batch-instance normalization, dropout, and techniques of this kind (including data augmentation). All with the objective to elucidate which regularization/normalization approach was better. As far as I understand, this experiment was not performed, could the authors explain why? ** AFTER READING RESPONSE ** After reading all reviewers' comments and the authors' feedback, I keep my original score. In my humble opinion, the authors' response is quite clear and useful, and I still think the paper represents, in general terms, a good submission. Empirical evidence suggests that the authors' proposal is a pertinent alternative to solve an important problem.

[Author Response · NeurIPS 2020]

# Author Response for "Stochastic Normalization"

We thank all reviewers for insightful and professional comments. In general, reviewers find the paper well written, the topic important, and the method novel. Major concerns are addressed here, which will be incorporated to the revision.

## Reviewer #1

**Question 1: Inconsistent experimental setup.** Some results were omitted due to space limit rather than cherry-picking. We agree with the reviewer that a consistent presentation of the results is important, and further present these detailed results in Table 1 and Fig. (a) below. In Fig. (a), the improvements over vanilla fine-tuning are averaged across three datasets to keep the plots simple. These consistent results confirm that StochNorm works well for a variety of backbones and datasets with different sampling rates. We will further provide the training/test splits of all datasets.

Table 1: Accuracy (%) of different methods on Chest X-ray, CUB-200-2011, and FGVC Aircraft datasets (backbone: ResNet-50).

| Method | Chest X-ray | | |
|---|---|---|---|
| | 5% | 10% | 15% |
| vanilla | 70.37 | 75.85 | 76.64 |
| $L^2$-SP | 70.02 | 72.73 | 75.71 |
| DELTA | 70.99 | 74.35 | 75.97 |
| BSS | 69.86 | 73.27 | 76.10 |
| StochNorm | 72.50 | 76.48 | 77.01 |

| Method | CUB-200-2011 | | | | FGVC Aircraft | | | |
|---|---|---|---|---|---|---|---|---|
| | 15% | 30% | 50% | 100% | 15% | 30% | 50% | 100% |
| $L^2$-SP | 45.08 | 57.78 | 69.47 | 78.44 | 39.27 | 57.12 | 67.46 | 80.98 |
| $L^2$-SP+**StochNorm** | 49.92 | 60.48 | 70.47 | 79.24 | 42.57 | 60.16 | 69.16 | 81.12 |
| DELTA | 46.83 | 60.37 | 71.38 | 78.63 | 42.16 | 58.60 | 68.51 | 80.44 |
| DELTA+**StochNorm** | 49.27 | 62.86 | 72.78 | 79.72 | 44.10 | 60.13 | 70.12 | 81.03 |
| BSS | 47.74 | 62.03 | 72.56 | 78.85 | 40.41 | 59.23 | 69.19 | 81.48 |
| BSS+**StochNorm** | 50.67 | 64.10 | 73.01 | 79.91 | 43.89 | 60.25 | 69.41 | 81.50 |

(a) Backbone · (b) Value of $\alpha$ · (c) Value of $p$ · (d) Compare with BIN

## Reviewer #2

**Question 1: The moving statistics hyperparameter $\alpha$ (PyTorch's $\alpha = 1 - $ Reviewer's $\beta$).** $\alpha$ is a hyperparameter of BatchNorm rather than of StochNorm. In **PyTorch**, $\alpha = 0.1$ by default. We are curious to the reviewer's question, and compare $\alpha = 0.1$ with $\alpha = 0.001$ in Fig. (b) above, showing the improvements over vanilla BN. Empirically we find that smaller $\alpha$ (larger $\beta$) leads to slower convergence and slightly worse results. A more comprehensive study on $\alpha$ will be included. We will clarify this detail as well as tone down the arguments related to Dropout.

**Question 2: Table 4 with other datasets.** Please see Table 1 above. StochNorm yields consistently better accuracies.

## Reviewer #3

**Question 1: How to avoid collapse when normalized by moving statistics.** This is an **insight** found this paper: in fine-tuning, collapse will occur only if **all** features are normalized by moving statistics. In StochNorm, some randomly selected channels are normalized by moving statistics while others by mini-batch statistics. This successfully stabilizes fine-tuning without relying on Batch Renormalization.

**Question 2: Ablation study of selection probability $p$.** As stated in the paper (Line 223), $p = 0.5$ works well for most experiments. Fig. (c) above presents an ablation study of $p$. Here $p = 1$ is BatchNorm, while $p = 0$ is to normalize **all** features with moving statistics (leads to collapse as pointed out by the reviewer). The best $p$ sits around $0.5$.

**Question 3: Over-claim the orthogonality.** Orthogonality means that StochNorm is architecture-based while L2-SP, DELTA and BSS are regularizer-based. In many cases they are complementary and an integration leads to better results. As such improvements are general but not absolute, we will tone down this claim.

**Question 4: Possible theoretical analysis.** Theoretical analyses for normalization are generally not easy. The papers of BatchNorm, GroupNorm, InstanceNorm neither provide theoretical analyses. Thanks for pointing it as future work.

## Reviewer #4

**Question 1: Comparison with other normalization approaches.** Fig. 3(b) in the paper compares StochNorm with BatchNorm (BN). There are no publicly available models pre-trained on ImageNet with instance normalization (IN) and weight normalization (WN). Only pre-trained models with BatchNorm are available. Batch-Instance Normalization (BIN) can be compared because it is based on BatchNorm. We present the absolute improvements over BN in Fig. (d) above, which shows that BIN is slightly better than BN but substantially inferior to StochNorm.

**Question 2: Comparison with data augmentation.** Data augmentations for all experiments (both StochNorm and other compared methods) are the same, as stated in supplementary material (Line 5). They are used by default in the computer vision community, and out of the scope of this paper, so we do not conduct ablations for data augmentations.

[Meta-Review · NeurIPS 2020]

Though the reviewers remark that the paper brings no insights/analysis, it was well-received by reviewers on average as an empirical architecture design idea, addressing an important problem. The experimental validation is conducted to the standards in the field and shows that the method is empirically useful. The combination BSS+StochNorm is particularly promising. The authors are invited to submit the final version, considering the following improvements: - the paper can be densified to avoid self-repetitions and redundancy (of definitions of normalizations, descriptions of the contribution -- something like trice, of the existing methods, the algorithm and its description and Fig 1) - this space and the 9th page could be used to clarify important details of the experimental setup that are needed to understand what is the basis of comparison: how the hyperparameters are chosen per method, whether the 5 trials include a random train-validation splitting; include additional results from the rebuttal and discuss more along along the points below relating to the literature. As pointed out by reviewers, using moving averages is princily different from using batch statistics in that the moving average is considered as a constant for back-propagation. The issues related to this should be highlighted and discussed more in the paper. In particular the claim that "BN reduces covariate shift" I believe was an imprecise statement in the original paper by Ioffe and is misinterpreted by many subsequent works in the field. It should mean that BN allows for a quick adjustment when a covariate shift happens. The empirical design idea to randomly mix parameter-sensitive batch statistics with parameter-insensitive moving averages brings something new that can motivate further research. - The following works should be discussed (more) as they are directly relevant: Ioffe, S. (2017) "Batch renormalization: Towards reducing minibatch dependence in batch-normalized models". because it combines batch statistics and running averages (deterministically with a schedule) Guo, etc. "Double Forward Propagation for Memorized Batch Normalization", 2018 a different variant related to the above Shekhovtsov and Flach (2018): "Stochastic Normalizations as Bayesian Learning" combine full-dataset statistics approximated analytically and introduce randomness to replicate the regularizing effect of BN Li et al. "Understanding the Disharmony between Dropout and Batch Normalization by Variance Shift" (2019) can be used to support arguments about dropout, as it directly studied the (in)compatibility of BN and dropout has been investigated If the discussion of Disout is relevant, please note that the journal paper on dropout also shows that a "gaussian dropout" with noise N(1,sigma^2) works equally well and thus already have shown a method that "perturbs neuron outputs rather than drops them" - A discussion and possibly some comparison with semi-supervised learning methods (as the experimental setup with 15% of data clearly allows this) would be important in order to see the larger picture. If I may, also a private request: the colorful hyperlinks (citations, sections, equations) are quite distractive, could you keep the hyperlinks but make them text black?